# Structuring of Bioceramics by Micro-Grinding for Dental Implant Applications

**DOI:** 10.3390/mi10050312

**Published:** 2019-05-09

**Authors:** Pablo Fook, Daniel Berger, Oltmann Riemer, Bernhard Karpuschewski

**Affiliations:** Laboratory for Precision Machining (LFM), Leibniz Institute for Materials Engineering (IWT), MAPEX Center for Materials and Processes, University of Bremen, 28359 Bremen, Germany; berger@iwt.uni-bremen.de (D.B.); riemer@iwt.uni-bremen.de (O.R.); karpu@iwt-bremen.de (B.K.)

**Keywords:** micro-grinding, bioceramics, materials characterisation, dental implant

## Abstract

Metallic implants were the only option for both medical and dental applications for decades. However, it has been reported that patients with metal implants can show allergic reactions. Consequently, technical ceramics have become an accessible material alternative due to their combination of biocompatibility and mechanical properties. Despite the recent developments in ductile mode machining, the micro-grinding of bioceramics can cause insufficient surface and subsurface integrity due to the inherent hardness and brittleness of these materials. This work aims to determine the influence on the surface and subsurface damage (SSD) of zirconia-based ceramics ground with diamond wheels of 10 mm diameter with a diamond grain size (d_g_) of 75 μm within eight grinding operations using a variation of the machining parameters, i.e., peripheral speed (v_c_), feed speed (v_f_), and depth of cut (a_e_). In this regard, dental thread structures were machined on fully sintered zirconia (ZrO_2_), alumina toughened zirconia (ATZ), and zirconia toughened alumina (ZTA) bioceramics. The ground workpieces were analysed through a scanning electron microscope (SEM), X-ray diffraction (XRD), and white light interferometry (WLI) to evaluate the microstructure, residual stresses, and surface roughness, respectively. Moreover, the grinding processes were monitored through forces measurement. Based on the machining parameters tested, the results showed that low peripheral speed (v_c_) and low depth of cut (a_e_) were the main conditions investigated to achieve the optimum surface integrity and the desired low grinding forces. Finally, the methodology proposed to investigate the surface integrity of the ground workpieces was helpful to understand the zirconia-based ceramics response under micro-grinding processes, as well as to set further machining parameters for dental implant threads.

## 1. Introduction

Dental implants aim to replace a partially or totally, damaged or diseased tooth structure, i.e., restoring the function and also the aesthetics [1,2]. In the last decades, this market has experienced growth and one of the solutions for dental treatment became the replacement of conventional metal-based dentures with ceramic materials [3]. The use of bioceramics is an alternate option to the toxic and allergic effects that might be caused by diffused metal ions due to corrosion and deterioration without wear of metal materials [4]. This alternative, however, is only possible because of the new developments in the field of biomaterials and computer-aided design/computer-aided manufacturing (CAD/CAM) technologies [5,6,7]. In this regard, the optimisation of CAD/CAM systems has enabled more efficient and cost-effective grinding processes in the scientific, industrial, and technological fields in a variety of sectors, such as aeronautics and biomedical [8].

Among the bioceramics in the market, aluminium oxide (alumina—Al_2_O_3_) and zirconium dioxide (zirconia—ZrO_2_) have become the better alternatives due to their combination of biocompatibility, mechanical properties, like high flexural strength and wear resistance, as well as minimum thermal and electrical conductivity [5,9,10]. In the specific case of zirconia, its phase transformation toughening phenomenon is known to improve the properties of the material. This phenomenon stops crack propagation, resulting from the transformation of zirconia from the tetragonal phase into the monoclinic phase, as well as the consequential 3% to 5% volume expansion and induction of compressive stresses. The interest in the toughening mechanics of zirconia allowed for the development of further zirconia-based ceramics, such as alumina-toughened zirconia (ATZ) and zirconia-toughened alumina (ZTA) [1,2,5,11].

The structuring of bioceramics using micro-grinding is still an area under investigation, and the surface integrity characterisation of ground ceramics is considered to be a key aspect of their further applications as dental implants on the market [1,2,3]. In this study, three types of fully sintered zirconia-based ceramics machined by micro-grinding were characterised by monitoring the process forces and measuring the surface integrity of the ground workpieces. The grinding strategy suggested that replicating the square thread profiles of dental implants using diamond galvanic-bonded wheels and optimising the machining parameters, i.e., peripheral speed (v_c_), feed rate (v_f_), and depth of cut (a_e_), using the design of experiments (DOE) method as the statistical approach for planning, conducting, analyzing and interpreting data from grinding experiments. The results evaluate the bioceramics with regard to their machinability and, thus, their suitability as materials for dental ceramics. 

## 2. Materials and Methods 

### 2.1. Case Study

The success of ceramic dental implants is correlated with effective osseointegration, such as the formation of direct contact between the implant and the surrounding bone [11,12]. According to the literature, to establish reliable osseointegration, six main factors should be considered: material selection, implant design, an optimum range of surface roughness, bone status, surgical technique, and loading conditions [13,14]. The last three points are correlated with the dentist’s expertise and biological factors concerning surgical planning and tooth restoration. Therefore, the first three factors are directly influenced by the grinding conditions [9,13,14].

In order to avoid defect parts during manufacturing and failures during its use, enhanced micro-grinding processes are still necessary and, consequently, have been subject to several research investigations [3,6,7,9,13,15,16,17,18,19]. Crucial requirements for high surface integrity and mechanical reliability of dental ceramic implants are knowledge and control of the critical machining parameters that are based on the materials, the implant overall design, and custom-designed requirement, i.e., the patients’ needs. Therefore, based on the current trends in dental implants, a wide variety of factors must be considered in threads design and component manufacturing. Figure 1 lists some of these characteristics according to the literature, as well as the relevant features and the ranges selected as optimal for the successful performance of the dental implants, namely shape, dimensions, surface roughness, and thread pattern. Specifically, implant threads are designed to maximize initial contact, provide primary stability, enhance the surface area, cause compression of bone, facilitate dissipation of loads at the bone-implant interface, and minimize the micro-movement to hazen osseointegration [12,13,20,21]. Multiple investigations have concluded that the square thread profile may provide the best primary stability and the most effective stress distribution in an immediate loading situation [20,21,22].

### 2.2. Material Selection

The ceramic workpieces (5.0 mm × 7.0 mm × 33.0 mm) are fully sintered and commercially-available tetragonal polycrystalline zirconia (ZrO_2_-TZP),—also known as zirconium dioxide (ZrO_2_) and commonly called ‘‘zirconia’’—alumina toughened zirconia (ATZ), and zirconia toughened alumina (ZTA). These materials are zirconium-based ceramics commonly used for dental applications and have intrinsic toughening mechanisms, but differ in their mechanical properties [5,23], as shown in Table 1.

### 2.3. Process Kinematics and Experimental Conditions

In order to machine dental threads, as illustrated in Figure 1, the micro-grinding process kinematics were carried out on a DMG Sauer 20 linear machine tool (DMG Sauer GmbH, Bielefeld, Germany) under a water-based lubricant that also provided cooling, lubrication, and chip removal. The machining of the ceramic workpieces was performed by a tool feed (v_f_) along the x-direction, while the tool spindle rotated (n_p_). In this case, diamond galvanic-bonded wheels of 10 mm in diameter, commercialised by the company SCHOTT Diamantwerkzeuge GmbH, with a specific width (b_w_) of 0.9 mm, and an average diamond grain size of 75 μm (D75) were used [24,25]. Moreover, the machining strategy and the diamond wheels used to grind the bioceramics were designed to follow the characteristic dental implant thread width (t_w_) of 0.2 mm, as well as the overall design of a square thread profile of a dental implant. Figure 2 illustrates the process kinematics.

The machining conditions performed are summarised in Table 2, i.e., peripheral speed (v_c_), feed speed (v_f_), and depth of cut (a_e_), which were selected after a screening campaign and a literature review concerning the critical depth of cut, as well as the equivalent chip thickness, to achieve ductile grinding mode machining [6,7,10,16,17,18,19,26]. 

In the present work, the Taguchi method was used as the design of experiment (DOE) approach to examine the influence of the grinding process parameters on the surface integrity and grinding forces of the three bioceramics materials [27,28,29,30]. As a result, eight process conditions (Table 3) were designed and every experiment (P1 to P8) was performed three times for statistical purposes. 

### 2.4. Workpiece Characterisation and Process Monitoring

In order to evaluate the surface and subsurface damage (SSD), the microstructure and surface topography were studied by means of a scanning electron tabletop microscope TM3030 (Hitachi Ltd., Hitachi, Japan) under magnifications of 60× and 2500×. An X-ray diffraction (XRD) machine (Bruker Co., Billerica, USA) was used to measure the residual stresses along the diagonal stress axis (Cu Kα-source, 30 kV, and 40 mA radiation). Finally, the surface roughness was measured by using a white light interferometer Talysurf CCI HD (WLI Taylor Hobson, Leicester, UK) within an air-conditioned laboratory. Herein, the roughness data was acquired using a Gaussian filter with a specified cut off λ_c_ of 0.08 mm and an objective of 50×.

Figure 3 depicts the force measurement system available at the DMG Sauer grinding machine. Herein, a three-component force dynamometer unit Kistler 9256-C2 (Kistler Holding AG, Winterthur, Switzerland) was used for the measurement of the grinding forces. A data acquisition and analysis software MesUSoft 2.5.23 (IWT, Bremen, Germany) was used for data collection and display. This study focused on the forces applied to the y-direction (Fy) as a methodology to monitor the machining process. The mean force (Fy) was estimated according to the average values of each grinding step in the y-axis.

## 3. Results

### 3.1. Surface and Subsurface Damage (SSD) Evaluation

#### 3.1.1. Microstructure

Figure 4 and Figure 5 show the microstructure modification of the ground threads as a result of the most representative process conditions investigated. Figure 4 corresponds to ground bioceramics machined with the highest possible parameters selected for this study (P1), while Figure 5 corresponds to the less demanding configuration (P8). 

The surface of the ground ZrO_2_ dental threads, with a 75-grit diamond wheel, at peripherical speed (v_c_) 10.00 m/s, feed rate (v_f_) 100 mm/min, and a depth of cut (a_e_) 50 μm, i.e., process condition P8, showed ductile streaks and a smooth surface as indicated in Figure 5. The same bioceramic ground at v_c_ 18.33 m/s, v_f_ 300 mm/min, and an a_e_ 250 μm, i.e., process condition P1, as shown in Figure 4, had a ductile area with micro-ploughing deformation. In both images, ZrO_2_ ground workpieces showed a greater amount of ductile areas than the ATZ and ZTA specimens, where the material was removed in a more partial ductile grinding and brittle mode. 

#### 3.1.2. Residual Stress

According to the XRD analysis, shown in Figure 6, compressive and tensile residual stresses of the ground samples were observed for P1 and P8. Grinding of the ZrO_2_ workpieces predominantly increased the compressive stresses, i.e., −141 MPa for P1 and −52 MPa for P8. Both machined ATZ samples exhibited tensile stresses after grinding, for instance, 178 MPa and 133 MPa for P1 and P8, respectively. For ZTA specimens, the P1 tended to generate a slightly tensile stress of 6 MPa and substantial compressive stress for P8, herein −92 MPa. This phenomenon was due to the toughening mechanism, which also involves the phase transformation already mentioned. In general, compressive values are, likewise, desired in the specimen surface for biomedical application. For example, the compressive residual stress tends to increase the fatigue strength and the fatigue life of ceramic dental implants [1,5,10,19].

#### 3.1.3. Surface Roughness

Figure 7 shows the surface roughness values, Sa (arithmetical mean height), of the as-received and ground ZrO_2_, ATZ, and ZTA specimens. Although the same process conditions were applied for all the three bioceramics, different surface roughness were achieved. The values for the ATZ dental threads were all considered to be the optimum results for the successful osseointegration of biomedical implants. For example, dental implants are suggested to exhibit a surface roughness, Sa, of between 500 and 1000 nm [1,20,22]. Therefore, for further use of ZTA materials as an implant, processes P2, P3, P4, P7, and P8 had an optimum range for the dental application; for ground ZrO_2_ ceramics, this was only observed for P3, P4, P7, and P8.

In general, the highest variation in Sa values, with respect to the eight machining conditions, was observed on ground ZrO_2_ workpieces, i.e., from ca. 715 nm up to 2100 nm. In comparison, ATZ and ZTA ceramics ranged between ca. 510 nm and 960 nm, and between ca. 820 nm and 1200 nm, respectively. 

### 3.2. Process Monitoring

#### Grinding Forces

Figure 8 shows the average values of each group of the bioceramic specimens and machining conditions, according to the methodology mentioned in Section 2.2. Due to the higher depths of cut in the first 4 processes (i.e., P1 to P4 and an a_e_ of 250 μm), higher mean forces (Fy) were also measured in comparison to the machining conditions that followed (i.e., P5 to P8 and an a_e_ of 50 μm). Specifically, for the machining of ATZ, higher forces were reported in regard to the grinding of ZrO_2_ and ZTA, which obtained similar values. In general, lower forces are beneficial for the tool wear and tool life [6,31].

## 4. Discussion 

In this section, further analysis of the surface and subsurface damage (SSD) of ground bioceramics threads are discussed. Moreover, the Taguchi method is used for understanding the mean Sa and F responses, based on the eight designed machining parameters [27,28,29,30].

### 4.1. Microstructure 

In Figure 4 and Figure 5, the surface of ATZ and ZTA workpieces show brittle intercrystalline breakouts, high roughness, and bulging at the scratch edges on the microstructure. The brittle outbreak marks are predominant in the ZTA specimens, which links to the higher hardness and lower flexural strength of the material in comparison to the ZrO_2_ and ATZ ceramics. This showed that brittle materials led to different surface topographies although grinding conditions did not vary.

The grinding direction is clearly discernible in both figures. The surfaces consist mostly of a series of parallel grinding marks, and the width of these forms are better visible in Figure 4 than Figure 5 due to the higher machining conditions selected. For that reason, process condition P8 tended to generate less defects and flaws on the microstructure surface of the zirconia-based ceramics.

An important characteristic of a ceramic dental implant is the ability to create correct interaction between the ceramic implant and the bone tissue through the ground threads [32]. Since most of the implant surface is in direct contact with bone tissue, form and integrity of the microstructure surface have a great influence on successful osseointegration. Herein, the ZrO_2_ ground ceramics have a tendency to have a higher osseointegration response once a better-machined surface is obtained than the ATZ and ZTA bioceramics [12,14,20,21,22,32].

### 4.2. Residual Stress 

According to the XRD results, the highest machining parameters (P1, i.e., v_c_ = 18.33 m/s, v_f_ = 300 mm/min, and a_e_ = 250 μm) were more beneficial for ZrO_2_ workpieces in comparison to the less demanding grinding conditions investigated (P8, i.e., v_c_ = 10.00 m/s, v_f_ = 100 mm/min, and a_e_ = 50 μm) once more compressive stress was measured. However, in the case of the ATZ specimens, condition P1 resulted in higher tensile stresses than P8. No significant stress differences were seen in ZTA ceramics ground with condition P1, but the machining parameter P8 generated considerable compressive stresses.

The different responses on the residual stresses of ZrO_2_, ATZ, and ZTA ceramics are a function of their different intrinsic physical properties, material processing, machining history, and zirconia phase amount of each material [5,9,10,11]. The amount of zirconia phase was substantially distinct among the three bioceramics investigated (Table 1). Consequently, the machining effect on the crystallographic structure transformation from the zirconia tetragonal phase into the monoclinic phase to induce compressive stresses were also different.

The phase transformation in ceramics is a combination of the kinetics of diffusion-controlled as well as of diffusionless transformations at different strain rate and contact zone temperature by a martensitic transformation that occurs in the zirconia phase where the crystal structure changes from a tetragonal to a monoclinic structure and generates a 3% to 5% volume expansion [10,11,33,34]. 

Compressive stresses are ideally better accommodated by the complete implant–prosthesis system since the cortical bone is stronger in compression and weaker under shear and tensile forces [1,10,11,19]. In practical situations, the total contact area between the implant and bone may apply shear stresses that are transferred along with the interface, which can be harmful to the jaw and even destructive to the implant if a denture is wrongly chosen. Hence, the ground ZrO_2_ ceramics were indicated as the best option due to the higher compressive residual stress observed in both grinding conditions analysed.

### 4.3. Influence of Processing Parameters on Surface Roughness and Grinding Forces

The Taguchi method was used as a statistical tool for the optimisation of the grinding process by analysing the machining parameters’ influence on surface roughness and grinding forces. Therefore, an analysis of variance (ANOVA) was performed and evaluated using the statistical software Minitab 17 [35]. ANOVA results were carried out by separating the total variability of each machining parameter and its error. The main machining factors, peripheral speed (v_c_), feed speed (v_f_), and depth of cut (a_e_), and their response on the surface roughness (Sa) and grinding forces (Fy) on the three bioceramics were analysed [27,28,29,30,35]. 

To examine the differences between the most and the least demanding machining parameters conditions (P1 and P8), the main effect plots were generated with Minitab 17 support [35]. Basically, there was a main effect response when the different levels of a grinding parameter affected the surface roughness as well as the mean forces in the y-axis differently. The main effect plot graphs were visualized by the response mean for each machining parameter connected by a line. When the line tended to be horizontal (parallel to the x-axis), there was no main effect. When the line was not horizontal, there was a main effect or influence between the two grinding parameters selected in relation to the response investigated—herein, the response to surface roughness (Sa) and forces (Fy) in the process. Therefore, different levels of the factor affected the response differently. The steeper the slope of the line, the larger the magnitude of the main effect.

#### 4.3.1. Surface Roughness

The individual grinding parameters effects, i.e., peripheral speed (v_c_), feed speed (v_f_), and depth of cut (a_e_), on the surface roughness of ZrO_2_, ATZ and ZTA specimens are presented in Figure 9. All the bioceramic materials showed a similar trend regarding the peripheral speed (v_c_), which had the highest influence on surface roughness. Feed speed (v_f_) and depth of cut (a_e_) had the lowest contribution factor, but different inclinations according to the tested material, i.e., the slope rose or fell with the increase of the respective machining parameter level tested.

For all three bioceramic materials, v_c_ had a very strong slope line, which indicated the highest influence on surface roughness (Sa). Therefore, increasing the rotation speed also led to an increase in surface roughness. The highest v_c_ response on surface roughness indicated that the setting of 10.00 m/s peripherical speed was beneficial to achieve an optimal Sa range, independent of the v_f_ and a_e_ designed in this study. This was the reason that the machining conditions P3, P4, P7, and P8 were indicated for grinding all three zirconia-based ceramics investigated once the Sa achieved between 500 and 1000 nm are an optimum range for further dental applications as also mentioned in Section 3.1.3. Surface Roughness [1,20,22].

Additionally, once the feed rate increased, the surface roughness decreased for the ZrO_2_ and ATZ ceramics and increased for ZTA. Although the a_e_ main effect line tended to be horizontal for the machined bioceramics and, consequently, no significant Sa response was observed, the increase in the depth of cut for grinding ZrO_2_ and ZTA seemed to have a slightly positive influence on the material surface, while the opposite was observed for ATZ specimens.

#### 4.3.2. Grinding Forces

The quality of the dental part produced by the micro-grinding process is influenced by the grinding tool and the conditions, which are linked in particular by the induced mechanical forces [2,6,10,29,33]. As a result, the normal forces (Fy) monitored for each bioceramic during the machining are based on the workpiece material properties and consequently chip formation and ploughing force [6,10,16,22,31].

Similar force values were monitored during the grinding of the ZrO_2_ and ZTA ceramics. Both materials showed a significant lower process force response than the ATZ materials. Moreover, the highest forces measured in the process conditions with the higher depth of cut, i.e., a_e_ = 250 μm, were not desired because of expected higher tool wear during machining.

The forces measured while machining of the ATZ workpieces were essentially two- to three-times higher than the other two bioceramics. Basically, the long term machining of the ATZ ceramics tended to introduce more damage to the grinding tool life and to the surface integrity of the implant in comparison to the ZrO_2_ and ZTA materials.

The parameter setting P3 (v_c_ = 10 m/s, v_f_ = 300 mm/min, and a_e_ = 250 μm) exhibited the highest forces monitored, while the lowest forces were seen with condition P6 (v_c_ = 18.33 m/s, v_f_ = 100 mm/min, and a_e_ = 50 μm). Therefore, in this study, high rotation, low feed speed, and low cutting depth tended to be beneficial to keep process forces low. This machining configuration is explained in Figure 10.

The Figure indicates the process forces response during the grinding. In general, the ZrO_2_, ATZ, and ZTA materials showed similar tendencies regarding the machining factors influences. The depth of cut (a_e_) had the highest contribution, and peripheral speed (v_c_) and feed speed (v_f_) had the lowest impact factor on the ceramics. 

For all bioceramics, a_e_ had a very strong line inclination, which points out the highest influence on the grinding forces (Fy) response once the cutting depth was increased. Furthermore, the same tendency with a lower slope was seen for the v_f_ main effect, where the force response also increased when the feed speed set was higher. The opposite configuration was seen once the peripheral speed rose. Herein, the forces applied to the ceramics decreased.

## 5. Conclusions

Three fully sintered types of zirconia-based ceramics, zirconium dioxide (ZrO_2_), alumina-toughened zirconia (ATZ), and zirconia-toughened alumina (ZTA) were structured by micro-grinding process. In order to replicate dental threads with a square profile, a grinding wheel with a diameter of 10 mm and a specific width (b_w_) of 0.9 (grain size: D75) was used. Eight machining conditions were designed and the process forces (F) were monitored. The microstructure of the ground bioceramics was analysed via SEM, the XRD technique accessed the residual stresses, and surface roughness (Sa) was measured with WLI.

The following conclusions were drawn from the investigation:The microstructures of the ground ATZ and ZTA workpieces showed brittle intercrystalline breakouts, high roughness, and bulging at the scratch edges. Although the ground ZrO_2_ surfaces had parallel grinding marks with micro-ploughing deformation, their microstructure had a larger amount of ductile areas than the other specimens.For a successful implant and mechanical stability in the jaw, compressive residual stresses on the material surface are recommended. ZrO_2_ ceramics had shown the best response concerning the residual stresses among the ceramics tested for dental application. Herein, higher compressive stresses after grinding were observed due to the toughening mechanics of the zirconia phase.The different surface roughness (Sa) and force (F) responses due to the different grinding parameters were directly correlated to the intrinsic physical properties and chemical composition of the bioceramics investigated. Basically, the machining conditions P3, P4, P7, and P8 generated the optimal surface roughness suggested for dental implants, i.e., between 500 and 1000 nm, on all machined bioceramic materials. This was due to the highest peripheral speed, v_c_, response on the surface roughness, which indicated that the low level tested (10.00 m/s) was beneficial to achieve an optimal Sa for dental uses. Regarding the process monitoring, the depth of cut (a_e_) had the highest influence on the grinding forces (Fy) response when it was larger—herein, the machining process with an a_e_ of 50 μm are indicated for less tool wear and best implant integrity.ZrO_2_ ceramics machined with the grinding conditions P7 (v_c_ = 10.00 m/s, v_f_ = 300 mm/min, and a_e_ = 50 μm) and P8 (v_c_ = 10.00 m/s, v_f_ = 100 mm/min, and a_e_ = 50 μm) are suggested for further dental applications due to their optimal Sa range, smoother microstructures, compressive residual stress, as well as low forces generated during machining.

Based on the results of this work, a future investigation should include a similar approach to machine aluminium oxide (alumina—Al_2_O_3_). This will extend the validity of this approach and will allow a fundamental material and process analysis in the micro-grinding of bioceramics. Finally, the next steps are the machining of ceramic dental parts based on the optimum grinding parameters investigated for further mechanical and biological evaluation.

## Figures and Tables

**Figure 1 micromachines-10-00312-f001:**
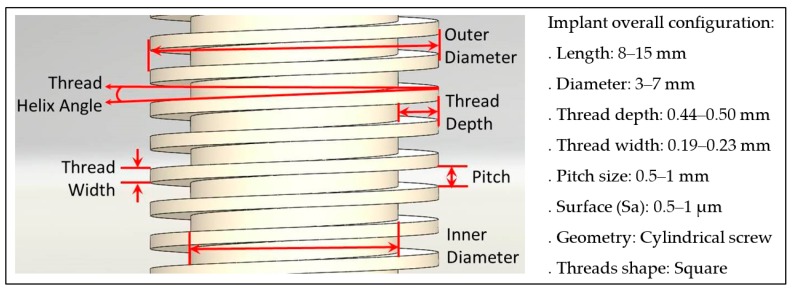
The overall design and characteristics of commercial dental implant threads [1,13,14,20,21,22].

**Figure 2 micromachines-10-00312-f002:**
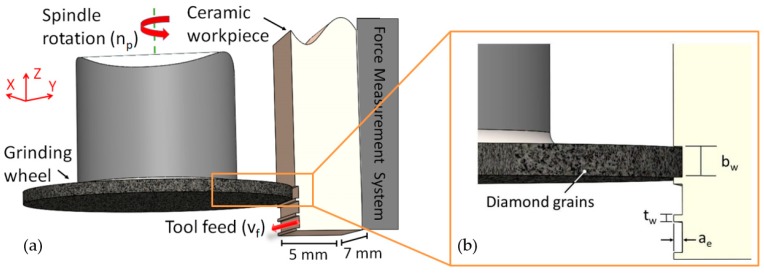
Illustration of the micro-grinding strategy performed in this work (**a**) in which the tool engagement and the workpiece features are highlighted (**b**).

**Figure 3 micromachines-10-00312-f003:**
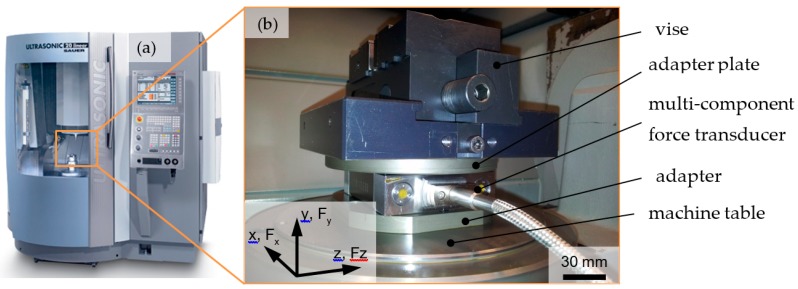
(**a**) DMG Sauer 20 linear machine tool and (**b**) setup for force measurement system.

**Figure 4 micromachines-10-00312-f004:**
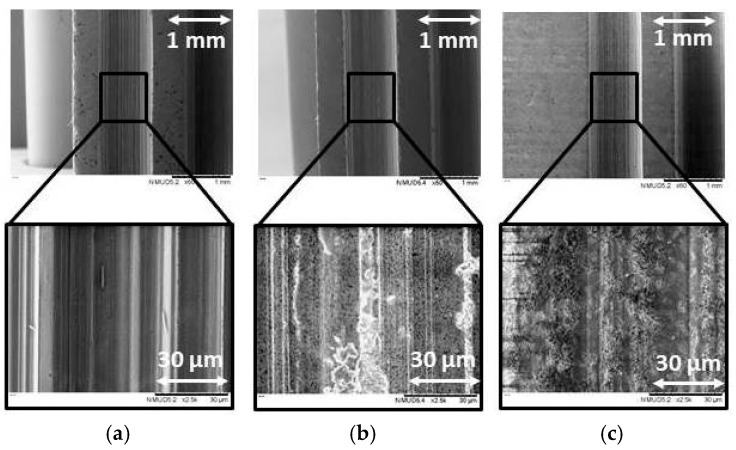
SEM analysis of the ground bioceramics under magnifications 60× and 2500×, i.e., for (**a**) ZrO_2_, (**b**) ATZ and (**c**) ZTA specimens. The bioceramics were machined with process condition P1, i.e., v_c_ = 18.33 m/s, v_f_ = 300 mm/min, and a_e_ = 250 μm.

**Figure 5 micromachines-10-00312-f005:**
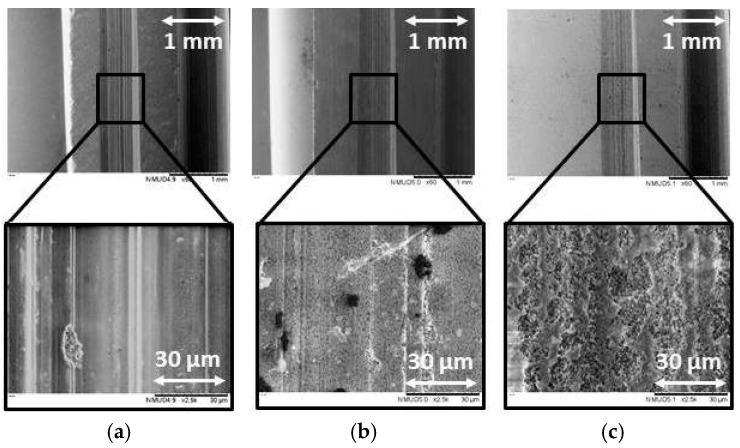
SEM analysis of the ground bioceramics under magnifications 60× and 2500×, i.e., for (**a**) ZrO_2_, (**b**) ATZ, and (**c**) ZTA specimens. The bioceramics were machined with process condition P8, i.e., v_c_ = 10.00 m/s, v_f_ = 100 mm/min, and a_e_ = 50 μm.

**Figure 6 micromachines-10-00312-f006:**
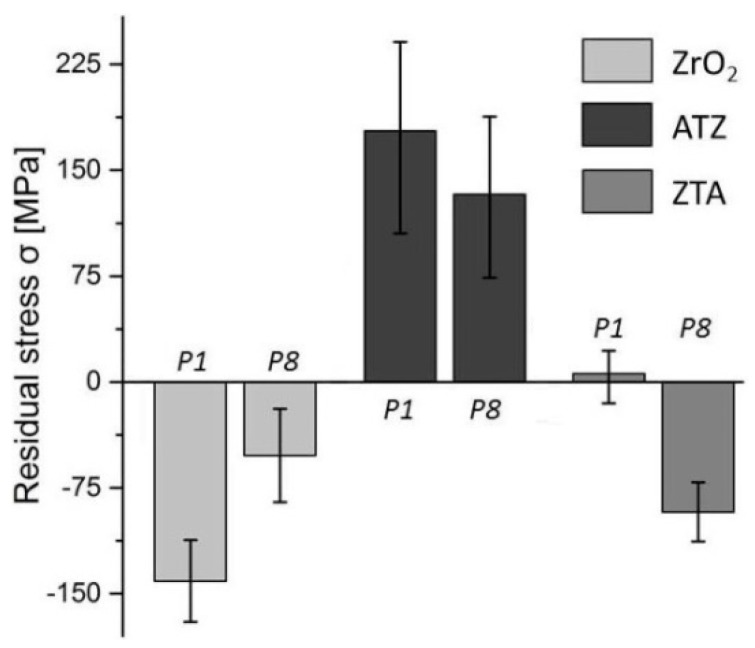
Residual stresses measured with X-ray diffraction technique of the ground ZrO_2_, ATZ, and ZTA ceramics machined with parameters P1 (v_c_ = 18.33 m/s; v_f_ = 300 mm/min; a_e_ = 250 μm) and P8 (v_c_ = 10.00 m/s; v_f_ = 100 mm/min; a_e_ = 50 μm).

**Figure 7 micromachines-10-00312-f007:**
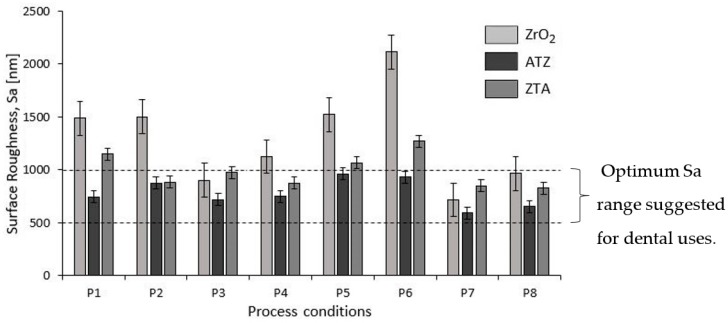
The surface roughness, Sa (nm), per material of the ground dental threads. (Table A1).

**Figure 8 micromachines-10-00312-f008:**
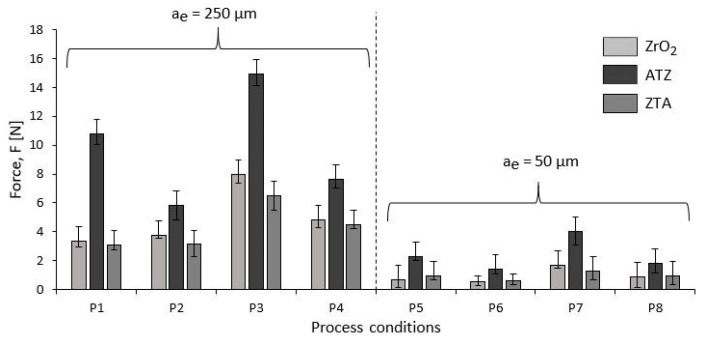
Mean forces Fy (N) for the grinding processes per material. (Table A2).

**Figure 9 micromachines-10-00312-f009:**
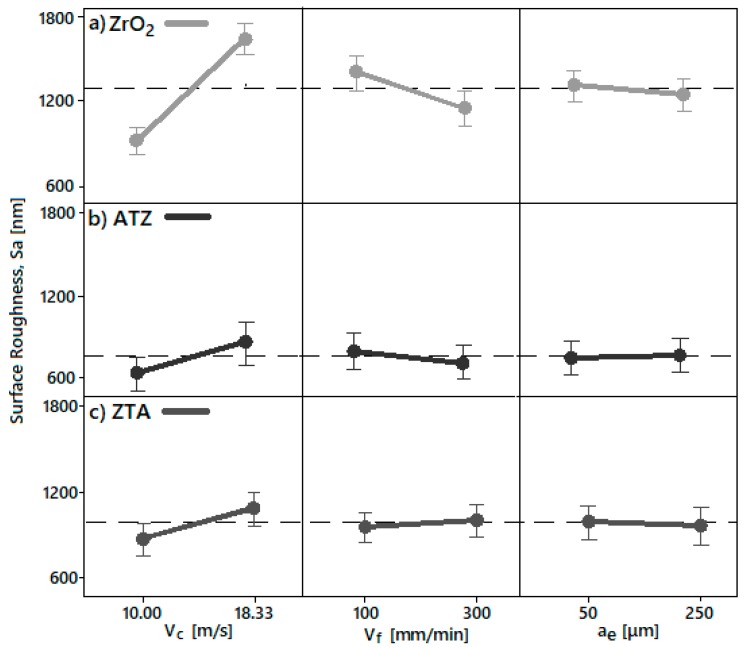
The main effect diagram for surface roughness (Sa) of (**a**) ZrO_2_, (**b**) ATZ, and (**c**) ZTA in regard to peripheral speed (v_c_), feed rate (v_f_), and depth of cut (a_e_).

**Figure 10 micromachines-10-00312-f010:**
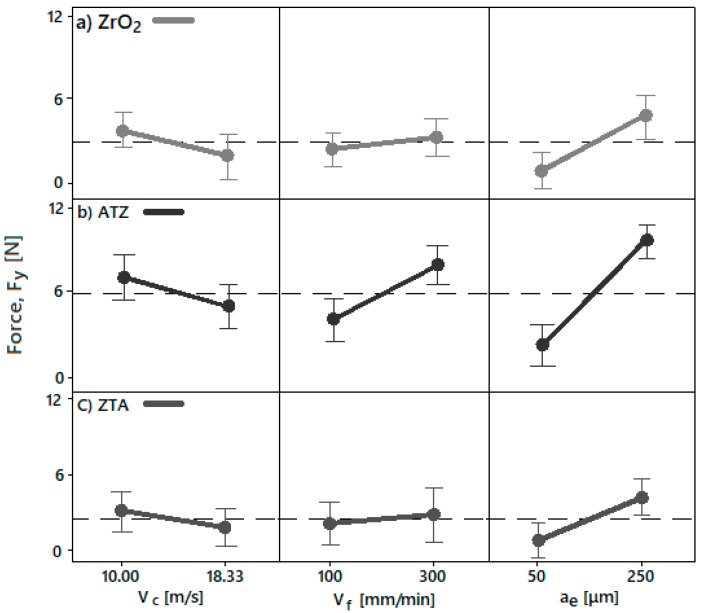
The main effect diagram for forces (Fy) of (**a**) ZrO_2_, (**b**) ATZ, and (**c**) ZTA in regard to peripheral speed (v_c_), feed rate (v_f_), and depth of cut (a_e_).

**Table 1 micromachines-10-00312-t001:** Description of the material properties [23].

Material	Density (g/cm^3^)	Fracture Toughness K_IC_ (MPa m^1/2^)	Young’s Modulus (GPa)	Hardness HV_10_ (GPa)	Flexural Strength (MPa)	Fraction of ZrO_2_ (%)
ZrO_2_-TZP	6.03	4.8	200	11	1000	> 95
ATZ	5.50	7	220	14	820	76
ZTA	4.10	8	380	16	440	14

**Table 2 micromachines-10-00312-t002:** Grinding conditions and material selection.

Peripheral Speed, v_c_ (m/s)	Feed Speed, v_f_ (mm/min)	Depth of Cut, a_e_ (μm)	Material
10.00, 18.33	100, 300	50, 250	ZrO_2_, ATZ, ZTA

**Table 3 micromachines-10-00312-t003:** Design of the process conditions in this study.

Process Condition	Peripheral Speed, v_c_ (m/s)	Feed Speed, v_f_ (mm/min)	Depth of Cut, a_e_ (μm)
P1	18.33	300	250
P2	18.33	100	250
P3	10.00	300	250
P4	10.00	100	250
P5	18.33	300	50
P6	18.33	100	50
P7	10.00	300	50
P8	10.00	100	50

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
