# Peer review of "Structuring of Bioceramics by Micro-Grinding for Dental Implant Applications"

_micromachines, 2019, doi:10.3390/mi10050312_

Round 1
Reviewer 1 Report
Paper needs several improvements.

Reviewer 2 Report
Acceptable with a minor spelling check.
Reviewer 3 Report
Topic of reviewed article is very important especially in context of complementary evaluation of quality of the ceramic workpiece material shaped by micro-grinding process. Reviewed article is interesting and write in high scientific level. Described problem is analyzed in complementary manner and scientific arguments were given. However the manuscript have some week points mentioned below:
the main findings of this study should be mentioned in the abstract,
the aim of the whole study should be precisely defined at the end of introduction section – lack of a precise reason for undertaking the research,
in phrase: “5.0 x 7.0 x 33.0 mm3” unit is incorrect,
all symbols should be write italics,
please provide information about expenditure of coolant during micro-grinding,
Fig. 2 has two parts – should be marked as a) and b) and also described in figure caption,
in grinding process is only one grinding force F which has three components in the x, y and z directions according to ISO standard – please correct throughout the text,
please use proper designation of force components,
please provide precise name of Sa parameter in Figs. 7 and 9-11,
on page 7 phrase “In this session...” palpably should be changed into “In this section...”,
please provide information about possible applications of obtained results in other micro-grinding processes.
Round 2
Reviewer 1 Report
Paper is OK